# Role of Cardiovascular Magnetic Resonance in the Management of Atrial Fibrillation: A Review

**DOI:** 10.3390/jimaging8110300

**Published:** 2022-11-04

**Authors:** Davide Tore, Riccardo Faletti, Andrea Biondo, Andrea Carisio, Fabio Giorgino, Ilenia Landolfi, Katia Rocco, Sara Salto, Ambra Santonocito, Federica Ullo, Matteo Anselmino, Paolo Fonio, Marco Gatti

**Affiliations:** 1Radiology Unit, Department of Surgical Sciences, University of Turin, Azienda Ospedaliero Universitaria (A.O.U.) Città della Salute e della Scienza di Torino, 10126 Turin, Italy; 2Department of Radiology, Humanitas Gradenigo Hospital, 10126 Turin, Italy; 3Division of Cardiology, Department of Medical Sciences, University of Turin, Azienda Ospedaliero Universitaria (A.O.U.) Città della Salute e della Scienza di Torino, 10126 Turin, Italy

**Keywords:** atrial fibrillation, cardiovascular magnetic resonance, catheter ablation, late gadolinium enhancement, strain

## Abstract

Atrial fibrillation (AF) is the most common arrhythmia, and its prevalence is growing with time. Since the introduction of catheter ablation procedures for the treatment of AF, cardiovascular magnetic resonance (CMR) has had an increasingly important role for the treatment of this pathology both in clinical practice and as a research tool to provide insight into the arrhythmic substrate. The most common applications of CMR for AF catheter ablation are the angiographic study of the pulmonary veins, the sizing of the left atrium (LA), and the evaluation of the left atrial appendage (LAA) for stroke risk assessment. Moreover, CMR may provide useful information about esophageal anatomical relationship to LA to prevent thermal injuries during ablation procedures. The use of late gadolinium enhancement (LGE) imaging allows to evaluate the burden of atrial fibrosis before the ablation procedure and to assess procedural induced scarring. Recently, the possibility to assess atrial function, strain, and the burden of cardiac adipose tissue with CMR has provided more elements for risk stratification and clinical decision making in the setting of catheter ablation planning of AF. The purpose of this review is to provide a comprehensive overview of the potential applications of CMR in the workup of ablation procedures for atrial fibrillation.

## 1. Introduction

Atrial fibrillation (AF) is the most frequent arrythmia with an increasing prevalence over time. In Europe, the number of AF cases is expected to increase from about 8.8 million in 2010 to 18 million in 2060 [1]. The current prevalence of AF is 59.6 cases per 100 person-years in men and 37.3 in women [2]. Clinical and public health implications of AF are significant due to the increasing burden of the disease and because this condition causes significant morbidity and mortality, as it is an important risk factor for stroke. Moreover, AF is related to a significant reduction in patients’ quality of life in case of symptomatic disease.

Management of this condition has evolved over time, in particular after the introduction of catheter ablation procedures for pulmonary veins isolation (PVI) [3] by Haissaguerre et al., which is even proposed as a first line treatment for such a condition [4].

In the last decades cardiovascular magnetic resonance (CMR) has had an increasingly important role as a research tool to investigate the pathological substrate of AF and also in routine clinical practice for procedural planning and risk assessment.

This article aims to provide a detailed overview of the applications of CMR in the setting of preprocedural workup for atrial fibrillation ablation treatments in current clinical practice and for research purpose.

## 2. Pathophysiology of Atrial Myopathy

Atrial fibrillation is a supraventricular cardiac tachyarrhythmia, and it represents the most prevalent arrhythmia in the world. Its prevalence is exponentially increasing with aging, reaching up to 8% in the elderly population and it is associated with considerable morbidity and mortality [5].

Several medical conditions are recognized risk factors for AF: arterial hypertension, cardiomyopathies, obstructive sleep apneas, valvular dysfunctions, obesity, and diabetes [6,7]. A clear predisposing factor for the development of AF is the presence of spontaneous atrial ectopic firing from a focus located in one of the pulmonary veins (PVs) [3]. Aberrant firing foci are frequently located in the PVs because of the presence of spontaneous ectopy originating from muscular sleeves, characterized by a reduced effective refractory period. Moreover, discontinuous conduction properties within the PVs might also provide a substrate for reentry circuits [8]. Additionally, ion channels in the PVs have peculiar properties compared to the rest of the cardiac muscle and, in particular, PVs have smaller inward-rectifier K^+^-current (IK1) and L-type Ca^2+^-current (ICa,L) and larger delayed-rectifier K^+^-currents, compared to other left atrial cells [5]. This causes a reduction in action-potential duration (APD), and it increases the probability of spontaneous ectopy due to delayed afterdepolarizations (DADs) making reentry more likely.

After decades of research on AF, the EHRA/HRS/APHRS/SOLAECE expert consensus document recognized in 2016 a new clinical and histopathological entity: atrial cardiomyopathy (ACM) [9].

ACM has been described as an atrial pathologic process in which the alteration of the architecture, structure, contractile, and electrophysiologic properties of the left atrium (LA) might lead to clinically significant manifestations.

The histological and pathophysiological EHRAS (which is the acronym for EHRA/HRS/APHRS/SOLAECE) classification categorizes ACM in four classes:Cardiomyocyte dysfunction, related with genetic diseases, lone AF, and diabetes;Fibrosis-related disease, associated with aging and smoking;Mixed cardiomyocyte and fibroblast dysfunction, which generally happens in heart failure and valvular diseases;Non-collagen deposition disease conditions, which encompass various disorders, such as infiltrative cardiomyopathies (amyloidosis, including isolated atrial amyloidosis, granulomatosis, iron overload, microscopic adipose deposition, etc).

Class 1 ACM is mainly caused by genetical abnormalities. To date more than 100 genes have been described as potential risk factors for AF [5]; a certain number of such mutations are shared with ventricular cardiomyopathies. The most important genes related with AF are KCNQ1 (encoding the alpha-subunit of IKs channel) [10], which is involved in type 1 Long QT Syndrome (LQTS) [11], and familial forms of AF, TTN (encoding titin), implicated in approximately 25% of familial dilated cardiomyopathies (DCM) [12] and Lamin A/C (LMNA), involved in 5–10% of DCMs and considered an arrhythmogenic gene related to arrhythmias (in particular, atrial ones), conduction diseases and sudden cardiac death (SCD) [13].

Class 2 ACM is a primary fibroblast dysfunction, resulting in extensive LA scarring, particularly abundant in relation to the patient’s clinical profile (for example smoking status and age) [14].

Class 3 ACM is characterized by a mixed dysfunction of both cardiomyocytes and fibroblasts. In this class, there is a contemporary interplay of several pathologic processes (LA volume and pressure overload, stretching of cardiomyocytes and atrial dilatation, increased electrical atrial excitability, anomalies in cardiomyocytes apoptosis and repair process, deposition of collagenous tissue, etc) ultimately leading to atrial scarring [10].

Class 4 ACM is due to primary non-collagen infiltration of the atrial interstitial matrix (with or without changes in cardiomyocyte structure and function) caused by different conditions, such as amyloidosis, granulomatosis, iron overload, microscopic adipose deposition, and inflammation [9,10].

AF is known to promote arrhythmia maintenance and to lead to a progressive atrial structural remodeling. Such pathologic processes are summarized in the classic statement “AF begets AF”; these aspects are an important issue, which limit the long-term efficacy of both pharmacological and catheter ablation therapies.

## 3. Role of CMR for AF Ablation Planning

### 3.1. Pulmonary Veins’ Anatomy

Pulmonary veins (PVs) drain blood that was oxygenated in the lungs to the LA. They originate from capillary vessels in the lungs, converging into larger vessels, the interlobar PVs, draining different segments of the pulmonary lobe. The segmental and lobar veins finally unite into the PV. Pulmonary veins enter in the posterior LA wall; the left veins are generally located in a more superior position compared to the right ones. Relevant surrounding anatomical structures are the tracheal bifurcation, the esophagus, and the descending tract of the thoracic aorta; such structures are located behind the posterior left atrial wall [15].

The typical PVs’ anatomy includes the presence of four individual PVs, two for each lung, that separately enter into the LA. The normal pattern of four PVs is present in about 40% of the general population [16]. Atypical but not pathologic anatomical variations of pulmonary veins are reported in 30–38% of the population and may involve right or left PVs [17]. On the left side, the most frequent anomaly is a common draining trunk instead of two separate veins entering the LA that, based on the venous channel’s length, is distinguished in long and short variations. The short left common trunk variant is reported in about 15% of people, and it is the most common PV anatomical variation. Right-sided variants tend to be rarer and more complex: the most frequent right-sided variation is the presence of a third accessory right PV draining the middle lobe. Different subgroups and classifications based on the relation of the accessory vein to the normal PVs are reported in the literature [18]. Entry of all PVs into a common venous confluence is usually pathologic and often associated with syndromes. Among these, we recognize total anomalous pulmonary venous connection (TAPVC) and partial anomalous pulmonary venous connection (PAPVC) [19].

Regardless of their course, the pulmonary veins are internally lined by a short (about 9 mm) layer of myocardium, which is often the site of origin of pathologic electrical activity in AF, being that the left superior PV is the focus in almost 50% of cases [20].

CMR with angiographic sequences is able to depict PVs’ anatomy, to provide accurate sizing of the veins and of their ostia and to identify anatomical variations (accessory PVs, common ostium) or abnormalities (anomalous pulmonary venous return). The number, location, and diameters of the pulmonary veins are very valuable data provided by the radiologist to the electrophysiologist for the planning of AF ablation. Examples of PVs’ variations are reported in Figure 1.

In order to properly size the PVs and their ostia, cross-sectional orthogonal views are used to measure maximal and minimal diameters; ostial areas are obtained by multiplanar reconstruction.

### 3.2. Left Atrial and Left Atrial Appendage Volume

Left atrial remodeling has been considered to limit the efficacy of ablation procedures for AF. In particular, the degree of atrial remodeling has been classically assessed with transthoracic echocardiography (TTE) by measuring LA diameters and volumes [21].

The measurement of the size of LA is crucial for risk stratification in patients with AF. LA enlargement has been recognized as an independent risk factor and predictor of stroke and death for patients affected by AF [22]. Moreover, several studies and meta-analysis recognized that LA enlargement represents an independent risk factor for arrhythmia recurrence after catheter ablation [23,24,25,26]. Normal values for LA volume (LAV) reported in the literature are 38 ± 11 mL/m^2^ [27]. A LAV superior to 145 mL (measured with computed tomography) has been reported to be a strong predictor of arrhythmia recurrence [28]. It has been demonstrated that indexed LAV (iLAV), which is LAV indexed to body surface area (BSA), represents an even more precise independent predictor of AF recurrence after catheter ablation [26].

Even if TTE is the standard method used in routinary clinical practice to assess LA size, CMR is considered the reference standard for cardiac volumes assessment, including LA volumes because steady-state free precession (SSFP) techniques, or angiographic 3D acquisitions, do not rely on geometrical assumptions [29].

Left atrial appendage volume (LAAV) has been evaluated as a predictor of procedural success for AF ablation procedures [24,30]. Similar to LAV, larger LAAV have been associated with arrhythmia recurrence after catheter ablation. Large LAAV and LAA orifice diameters are also a risk factor for cerebrovascular events in AF patients [22].

### 3.3. Atrial Appendage Morphology and Stroke Risk

The presence of a continuous pathological electrical activity in the LA due to the constitution of functional macro- and micro-reentrant circuits determines the asynchronous contraction of the cardiomyocytes in the atrium. As a consequence, the atrial systole becomes inconsistent creating the conditions for the development of one of the main complications of AF, represented by thrombus formation in the LA and, more frequently, in the left atrial appendage (LAA) [31]. Thrombus formation may lead to subsequent strokes/TIAs and/or peripheral embolism in patients with AF.

LAA rests anteriorly in the atrioventricular sulcus in close vicinity to the circumflex coronary artery and the ipsilateral phrenic nerve and PVs. In an adult human heart, the LAA has a hemodynamic and neurohormonal regulatory function. As a remnant of the primordial LA, its shape is widely variable but is reducible to a long tubular hooked structure with different lobes. Its morphology is classically categorized in the literature in four main types by the number and placement of lobes, leading to increasing complexity [32]:chicken wing: single lobe, bent in the proximal/middle portion;cactus: central dominant lobe with secondary lobes extending omnidirectionally;windsock: bent dominant lobe, plus secondary or tertiary lobes;cauliflower: lack of a central dominant lobe with complex internal characteristics.

Examples of different LAA morphology are shown in Figure 2.

From a physiopathological point of view thrombus formation and thromboembolic events in patients affected by AF are caused by the classic Virchow’s triad (i.e., endothelial dysfunction, abnormal blood stasis, and hypercoagulable states). Blood stasis in the LAA, a fundamental thrombus catalyzer, could be fostered by higher volumes and sepimental complexity.

Given the improvements in image quality of CMR techniques and their increasing territorial availability, there has been great interest in the last decade in morphologically evaluating the LAA in order to better estimate thromboembolic risk, since stroke events are not that unlikely even in presumed low-risk patients (e.g., CHA2DS2-VASc score = 0–1) [33].

Studies have shown a correlation between LAA morphology and increased stroke risk in AF patients. In particular, the more complex the LAA morphology is, the higher is the risk of stroke, silent cerebral ischemic events, and TIAs. Non–chicken wing LAA morphologies have a significantly higher risk for an embolic event even in patients with low CHA2DS2-VASc scores [31,33,34]. CMR has a role in risk stratification and clinical decision making for AF patients at low risk of thromboembolic events to improve anticoagulation management and reduce their stroke risk as much as possible.

### 3.4. Atrial Appendage Occlusion

LAA occlusion with percutaneous devices is a feasible and an effective treatment for patients with AF who have contraindications for long term anticoagulant therapy.

Prior to LAA occluder device deployment, it is of the utmost importance to have an exact description of the LAA shape and accurate diameter measurement. LAA morphology, the depth, number, and position of its lobes are accurately depicted with imaging exams. Multi-plane reconstruction images are used to measure the diameter of the LAA ostium and landing zone. Device undersizing is associated with inadequate LAA blockage with residual peri-device blood flow with subsequent stroke risk, as well as device movement, dislodgement, or peripheral embolization [29].

CMR can give a comprehensive LAA assessment. However, there are no studies investigating its role in LAA appendage closure; this is likely due to the higher spatial resolution, scanning speed, and broader availability of CT scanners [35].

In clinical practice, LAA evaluation before percutaneous occlusion is usually performed with CT even if CMR may be a feasible alternative in selected patients.

### 3.5. Atrial Thrombus Assessment

Blood flow stagnation in the LA (in particular, in the LAA blind pouch) and impaired atrial contractility are the major causes of thrombus formation and possible consequent thromboembolism in patients affected by AF.

Before performing cardioversion (either pharmacological or electrical) and before catheter ablation procedures, it is imperative to exclude the presence of LA/LAA thrombi. TEE is usually performed to rule out the presence of thrombus in such locations. TEE is generally considered the gold standard for the detection of thrombi in the LA/LAA because of its elevated diagnostic accuracy [35].

CMR is a validated and effective tool for thrombus detection in the left ventricle and such technique is becoming the imaging modality of choice to evaluate the presence of left ventricular thrombi.

To our best knowledge, only few studies evaluated the possible role of CMR to detect LA/LAA thrombus. On cine images, the thrombus should appear as a mass inside the LA/LAA with distinct margins from the LA/LAA walls, and it should be distinguished from the pectinate muscles or artifacts. On contrast enhanced CMR angiography sequences, the thrombus should appear hypointense with margins discrete from the blood pool or from surrounding structures. On LGE-CMR sequences, LA/LAA thrombus should appear as a homogenous low signal intensity filling defect surrounded by higher signal blood pool [36].

### 3.6. Esophagus and Esophageal Thermal Injury

Esophageal thermal injury (ETI) is one of the worst complications of atrial radiofrequency ablation.

According to the literature, ETI can occur in up to 40% of cases [37], and it is probably caused by the ablation energy causing ischemia due to occlusion of the arterioles or damage to the vagal plexuses [38]. In more severe forms (<0.1%), it can cause transmural necrosis that may cause atrio-esophageal fistula (AEF) and mediastinitis, which is associated with a high mortality rate (55–85%).

Therefore, to prevent ETI, proper pre-procedural planning with imaging and the use of dedicate systems to protect the esophagus are mandatory.

Magnetic resonance is a standard imaging modality used for pre-ablation planning and can be useful to identify the position and course of the esophagus.

The anatomical vicinity of the esophagus to the surrounding structures has a strong impact on ETI prevalence, and three anatomical factors have been identified as potential risk factors [39]: the angle between the posterior wall of the esophagus and the descending aorta (LA-Ao angle), the angle of branching of the left inferior PV in the coronal plane (LIPV angle), and the distance from the posterior LA wall to the anterior surface of the descending aorta (LA-Ao distance). If the LA-Ao angle is small, the area of contact between the posterior wall of the LA and the esophagus could be larger, as well as if the LIPV angle is larger, and the LA-Ao distance is minor because of the proximity between the posterior wall of the LA and the esophagus increases.

Different techniques have been described to improve the visualization of the esophagus [40,41]. The use of a bolus of gadolinium-based contrast medium mixed with thickened water gel better depicts the esophagus true anatomical shape and size compared to baryta meal [40], and unlike this, it contains no gluten, lactose, or sugar; moreover, it is not adsorbed, and it is excreted with the feces at 99.2%. A Gd-BOPTA oral formulation was also tried because its signal intensity in T1-weighted sequences is greater compared to gadopentetate dimeglumine [40].

A viable alternative to using gadolinium-based contrast agents can be pineapple juice [41], after a proper concentration process and the addition of modified potato starch, as it is as hyperintense on T1-weighted images as MR-diluted contrast agents and allows feasible visualization of the esophagus without side effects. Examples of CMR images depicting the course of the esophagus and its anatomical relationship with LA structures are reported in Figure 3.

A study [42] demonstrated as a predictor of esophageal motion the minimal GAP between the posterior LA wall and the anterior vertebral body margin, which can be measured on axial scans during pre-operative planning. In particular, a gap greater than 4.5 mm should be predicted as a greater likelihood of esophageal motion than 10 mm; the study also showed that the movement of the esophagus could also depend on the amount of peri-esophageal connective tissue forming the “aorto-esophageal ligament”, which correlates with a higher BMI.

Different studies [37,43] have confirmed the high diagnostic accuracy of late gadolinium enhancement CMR (LGE-CMR) to detect ETI and assessing its extent and progression. The severity of ETI evaluated with LGE-CMR can be classified as none, mild, minimal, moderate, and severe. All cases of ETI confirmed by esophagogastroduodenoscopy (EGD) presented moderate or severe LGE on CMR. Compared to EGD, CMR could provide a more detailed characterization of the wall of the esophagus, showing edema on double inversion recovery (IR) T2 weighted turbo spin-echo (TSE) images and enhancement on LGE-CMR images, suggestive for wall flogosis, probably an early sign ETI. The study also demonstrated that on repeated LGE-CMR scans the severity of ETI significantly decreases reflecting the resolution of the transient esophageal lesion.

The most used CMR sequences used in the setting of pre-procedural planning of catheter ablation of AF and for post-procedural imaging are reported in Table 1.

## 4. Prognostic Value of Atrial LGE

### 4.1. Preprocedural LGE

Fibrosis is a multifactorial phenomenon that is generally a consequence of myocardial infarction, arrhythmias, such as AF, idiopathic dilated cardiomyopathy and heart disease caused by systemic illness, such as hypertension, and diabetes [44,45]. These conditions cause continuous damage to the atrial wall; when, at a certain point, the damages cannot be properly repaired anymore, a remodeling process starts that deals with the alteration of the extracellular matrix composition [44,45], resulting in the deposition of collagenous cicatricial tissue.

There is a consistent number of works about the relevance of the correlation between the amount of heart fibrosis related to the probability of recurrence of AF after catheter ablation procedure. Catheter ablation is an important therapeutic option to restore sinus rhythm in patients with symptomatic AF. Some studies demonstrate that the unsuccessful results of the procedure may be related to the presence of an excess of atrial wall remodeling [46].

Left atrial fibrosis extension has been classified by the University of Utah Atrial Fibrillation LGE-MRI-based staging system in four stages depending on the amount of atrial enhancement [47]:Stage I, 0–5%;Stage II, 5–20%;Stage III, 20–30%;Stage IV, >30%.

It has been established that the probability of arrythmia recurrence is proportional to the percentage of fibrosis [46,47,48,49]. It is not completely clear whether fibrosis is a cause or a consequence of AF, but it has been demonstrated that a higher Utah stage of fibrosis corresponds to a higher risk of recurrence post-ablation; in particular, a 45% increased risk of AF recurrence for every 10% increase in atrial fibrosis has been observed [2].

Up to 70% of the patients with a Utah stage IV will experience recurrent atrial arrhythmia during the first year after the procedure, but this happens just up to the 20% of the patients with stage I [45,46,47,48,50].

Fibrosis is an independent AF recurrence risk factor also in the long term; in a five-year study, it has been shown that there is a more elevated risk of AF recurrence and multiple ablation procedures needed in patients with a higher degree of wall remodeling [46]. Others risks factors of AF recurrence are increased LA volume and diabetes, even if they are not as significant as the fibrosis extent is.

LGE-CMR is an excellent tool to determine the amount of fibrosis of the LA wall; it has been shown that LGE-CMR allows to assess and quantify in a relatively reproducible way the areas of wall structural remodeling (SRM).

LGE-CMR images are acquired 10–30 min after gadolinium-based contrast agent injection using 3D electrocardiograph (ECG)-gated acquisition with respiratory navigated inversion recovery (IR) sequences. Recently, T1 mapping imaging has emerged as an alternative quantitative technique for the assessment of LA fibrosis; such approach, however, has not yet been broadly adopted.

The use of LGE-CMR for the staging of atrial fibrosis in the setting of preprocedural exams for catheter ablation may provide useful information for clinical decision making to provide a tailored approach by identifying the patients that are likely to benefit the most from ablation procedures [47]. Example of staging of atrial fibrosis on dedicate software is shown in Figure 4.

The 2020 ESC guidelines provide a class IIa level C recommendation for structured characterization of AF patients encompassing risk assessment for stroke, symptom status, AF burden, and the evaluation of the substrate severity, which can be obtained with LGE-CMR [6]. Guidelines do not suggest a precise timing for atrial fibrosis quantification with LGE-CMR; in the DECAAF II trial, with is the largest study employing LGE-CMR in the pre-ablation setting, LGE imaging was performed approximately 30 days before the ablation [51].

LGE evaluation of atrial fibrosis, despite having been introduced several years ago [52], has not become part of routine clinical practice because of some concerns over its reproducibility and image processing techniques. The descriptions of the reproducibility results of LGE-CMR for the assessment of atrial fibrosis are extremely variable. Some centers show near-perfect agreement between observers [53,54,55], while others find low agreement between observers and between scans [56]. In addition, it has been demonstrated that changing the threshold at which signal intensity is classified as fibrosis has a substantial impact on fibrosis burden evaluation [57]. Sources of variation may include imaging parameters used for the acquisition, observer expertise, and the software employed for image analysis.

### 4.2. Procedure Induced Ablation Scars

The cornerstone of catheter AF ablation is complete electrical isolation of all PVs [58]. There are three main different percutaneous ablation techniques: radiofrequency (RF) ablation, cryoablation, and laser ablation [59]. RF energy uses 300–1000 KHz frequencies; cryocatheters deliver cryoenergy, and laser ablation uses a diode source to produce a beam with a wavelength of 980 nm [60]. All ablative techniques induce lesions to the atrial wall causing a repair process that results in the deposition of collagenous scar tissue. As collagenous scar tissue is electrically inert, it is unable to propagate pathological impulses from the PVs, which are frequently the trigger for AF.

Despite the effectiveness and variety of techniques, recurrence of atrial arrhythmias is frequent [61]. Recurrence is defined as a 30 s atrial arrhythmia of any kind (AF, tachycardia, or flutter), detected after a blanking period of 90 day, according to the HRS/EHRA/ECAS consensus document [61].

The presence of complete circumferential PVs’ ostial scarring demonstrated with LGE-CMR has been reported to be relatively rare after RF ablation procedures, and it has been associated with better clinical outcomes [62]. Ablation induced scar burden has been related to the risk of recurrence and, in particular, smaller atrial LGE areas are related to poor scar formation and worse outcome [63]. Examples of post-procedural LGE-CMR controls are shown in Figure 5 and Figure 6.

The prospective multicenter study DECAAF has shown that baseline fibrosis, ablation-induced iatrogenic scars, complete PVs’ encirclement, and residual fibrosis were associated with arrhythmia recurrence [47]. For these reasons, to evaluate and quantify post-ablation scars and residual fibrosis with LGE-CMR after ablation procedures may provide information for arrhythmia recurrence risk stratification. Residual fibrosis, defined as the remaining areas of atrial fibrous tissue that has not been modified by scars, is given by the entity of the overlapping between procedural induced ablation scars and baseline fibrosis (i.e., residual fibrosis = baseline fibrosis—scar overlapping with preexisting fibrosis) [47]. It was found that high residual fibrosis increases the risk of recurrent arrhythmia as incomplete PV encirclement [64].

Ablation scar gaps, defined as discontinuous lines of scars around PVs after initial ablation, visualized and quantified by LGE-CMR, remain the first cause of macro-reentrant atrial tachycardias (ATs) [65,66,67]. In this case, it was found that LGE-CMR can not only quantify ablation scars but also provide the target, through the gaps, to treat during redo ablation procedures. Thereby, homogenizing ablation induced scars from previous procedures can suppress recurrent atrial arrhythmias, and it may increase arrhythmia free survival [68]. Even if baseline atrial fibrosis remains an important predictor of arrhythmia recurrence, it is crucial to emphasize the role of substrate modification in AF catheter ablation as guide for redo ablation procedures.

## 5. CMR-Guided Ablation

Ablation procedures for AF are usually performed with hybrid guidance of fluoroscopic and electroanatomical imaging also taking advantage of fusion imaging with preprocedural CMR images. Even in experienced hands, such procedures expose the patient to a consistent radiation dose [69]. Preprocedural imaging with CMR may help the electrophysiologist to reduce procedural duration and radiation dose to the patient and to the operators and to avoid possible complications, such as ETI [69,70,71]. The integration of cardiac imaging into electroanatomical maps has represented an important improvement for ablation procedures providing insights about patient anatomy thus leading to more effective PVIs and increasing procedural safety. For these reasons, pre-procedural CMR imaging has become part of clinical practice to guide AF ablation. The use of CMR images in the electrophysiology lab is a multistep process:The DICOM images are imported in a dedicated workstation, and they are segmented;During the procedure the LA is electroanatomically mapped using a dedicated catheter;The anatomic CMR map is merged with the electroanatomic one to obtain a hybrid map; andFinally, the ablation catheter is navigated in the LA using a hybrid map [72].

Imaging guide for catheter ablation of AF has been demonstrated to provide better results compared to traditional electrophysiologically only guided procedures with significantly lower rates of arrhythmia recurrence [72]. Sample images of hybrid imaging are reported in Figure 7.

The lack of a generally accepted definition for pathologic substrate-based ablation technique probably is one of the reasons for the relatively low success and quite high recurrence rate of AF after the first procedure and need for subsequent redo ablations. As low-voltage atrial areas detected by electrophysiological mapping correlate quite well with structural alterations on LGE-CMR imaging the use and integration of LGE imaging into current clinical practice may provide a more reproducible and less operator dependent tool to quantify LA scarring and remodeling and to guide ablation thus improving procedural success rates. The use of a LGE-guided approach to ablation may be used to target fibrosis for a tailored approach to the arrhythmic substrate.

Targeting structural remodeling of the LA during ablation in an effort to increase catheter ablation outcomes is a relatively new strategy that appears to be promising. In patients with persistent AF, the identification of low voltage areas representing fibrotic LA tissue and subsequent ablation led to longer arrhythmia-free survival intervals compared to standard catheter ablation [56,73,74].

Using CMR guidance to identify arrhythmogenic foci in AF patients appears to be a highly promising option.

The DECAAF II trial, a multicenter, prospective, randomized trial of patients with persistent AF, used LGE-CMR-defined LA fibrosis as the target of the ablation procedures and compared the outcome of routine PVI procedures to ablations based on imaging-defined atrial fibrotic burden. No significant difference in terms of AF recurrence emerged between routine the PVI group and the group with LGE-CMR-guided ablation [51].

It is intriguing to consider that image-guided treatment of atrial fibrillation could be administered without radiation exposure using real-time interventional cardiac magnetic resonance (iCMR) guidance. Real-time iCMR guidance systems with actively tracked catheters and filtered local electrograms have been recently developed. One of the advantages of iCMR over traditional ablation procedures in cath lab is the possibility to eliminate registration errors, thus permitting to target arrhythmia substrate more precisely. Moreover, lesion formation may be visualized in real time using MR-thermometry techniques or immediately after the procedure with LGE imaging, thus permitting to promptly identify and treat ablation gaps.

There is a strong correlation between LGE areas on CMR and electroanatomical mapping, and it is feasible to conduct studies of electrophysiology with real-time iCMR, according to preclinical research [75,76,77,78]. Successful ablation of simple atrial arrhythmias, such as atrial flutter, using an early-generation iCMR-conditional ablation system and passively visualized catheters has been described in some case series. Electrophysiologists should be able to treat more complex arrhythmias, including AF and ventricular tachycardia, with next generation systems providing more accurate catheter tracking via active visualization, using fusion imaging based on 3D electroanatomic maps, LGE imaging and real-time iCMR [79,80].

In conjunction with CMR-apparent devices, real-time CMR may provide a comprehensive imaging solution for interventional cardiovascular procedures.

## 6. Atrial Function, Stiffness, and Strain

Structural remodeling caused by AF results in LA fibrosis, which may cause LA stiffening (LAS) and an impairment in LA contractile function [81].

LA function can be measured with CMR, which is the reference standard for the assessment of cardiac function, using SSFP cine images.

A reduction in LA ejection fraction (LAEF) has been recognized as a risk factor for cerebrovascular events in patients with AF, thus making LAEF an additional element for risk stratification in such a population [82].

Atrial stiffness has been associated in several studies with the presence of LA fibrosis: an increased LA scar burden has been associated with an increase in LAS, reflecting a deteriorated atrial reservoir function [83]. LAS is an important determinant of cardiac pump function: an increase in LAS results in reduced atrial stroke volume and forward blood flow [84]. Moreover, an increase in LAS may have an important role in the genesis, development, and perpetuation of AF. Patients with paroxysmal AF present an increased LAS, according to recent research [85,86]. Due to these facts, pre-ablation LAS evaluation may have a role as a predictor of AF recurrence [81].

LA deformation can be directly assessed with LA strain (AS), reflecting its electromechanical and functional integrity. AF patients present lower strain values compared to healthy controls, moreover global AS correlated with higher LAEF, and lower LA volumes [87]. It has been reported that atrial areas with LGE on IR images present lower values of regional AS compared to healthy ones. In individuals affected by AF, regional LA dysfunction is proportional to the extension of underlying fibrotic areas evaluated by LGE-CMR [87].

Reduced AS in patients with heart failure has been described as a predictor of new-onset AF [88]. Alterations in AS have been described in patients with cryptogenic stroke who subsequently developed AF, representing a new element for cerebrovascular events risk stratification [89].

AS may be a prognostic factor for ablation procedure as patients with a greater AS rate undergoing catheter ablation are less likely to experience arrhythmia recurrence.

LA strain may also be applied to evaluate LAS. The LAS index, a unique measure of LA diastolic function, increases with age, and it is higher in patients with chronic AF and in those who have undergone different catheter ablation procedures. Greater LAS index was independently linked with AF recurrence after ablation [85].

## 7. Cardiac Adipose Tissue

Obesity is a major risk factor for AF and a global epidemic [90]. Recently, there has been a growing interest on the role of cardiac adipose tissue in the development of AF, notably epicardial adipose tissue (EAT) and pericardial adipose tissue (PAT). EAT resides between the visceral pericardium and epicardial layer of the myocardium. Beyond the parietal pericardium is the location of PAT. EAT has higher biological activity compared to PAT. PAT and EAT wrap around the PVs as they enter in the LA. Multiple studies have shown a link between cardiac adipose tissue and AF. Even if the majority of works in the literature did not differentiate between PAT and EAT, there is evidence that fat deposits in direct contact with the LA myocardium (EAT) are more prone to contribute to the genesis of AF [91]. EAT paracrine properties and its release of adipo-fibrokines, molecules with pro-inflammatory and pro-fibrotic effects, contribute to atrial inflammation and fibrosis, thus leading to the genesis of the proarrhythmic substrate. Greater volumes or thicknesses of PAT were related with a higher prevalence of paroxysmal and chronic AF [91].

Various studies have demonstrated that PAT burden could be a predictor of AF recurrence following catheter ablation [92,93,94]. The evaluation of LA periatrial fat might have a role for patient selection and risk stratification to improve AF ablation outcomes.

CMR allows noninvasive assessment of EAT and PAT volumes. Sequences of fat–water separation (such as Dixon) permit to quantify the volume of adipose tissue. An example of periatrial fat segmentation is shown in Figure 8.

The intra-atrial fat infiltration, evaluated by a fat–water separated sequence, has been demonstrated to be a risk factor for the development of AF [95].

In a prospective investigation of LA-epicardial fat (LA-EAT) using 3D Dixon technique, it was found that patients with AF had significantly more LA-EAT than a control group with different cardiovascular diseases [96].

CMR imaging to evaluate cardiac adipose tissue may have a role in identifying individuals at risk of developing AF and to stratify the risk of arrhythmia recurrence in patients undergoing catheter ablation.

## 8. Conclusions

CMR can provide information to make catheter ablation procedures of AF safer and faster. Moreover, it can provide a one-stop-shop in the setting of preprocedural planning for an accurate risk stratification of stroke and arrhythmia recurrence. An accurate patient selection based on CMR data may improve the outcomes of ablation procedures and might be useful in selecting patients who are likely to benefit the most from such treatment, avoiding unnecessary procedures and reducing related costs. CMR has also provided insights into the arrhythmic substrate of AF and several physiopathological mechanisms of this condition.

## Figures and Tables

**Figure 1 jimaging-08-00300-f001:**
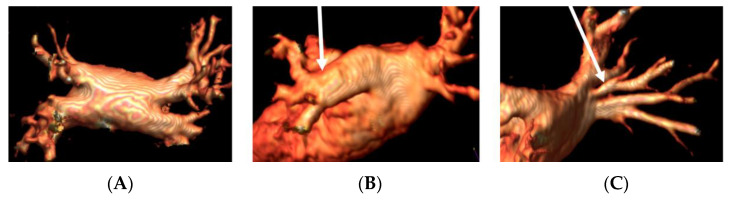
Panel (**A**), normal pulmonary veins anatomy; panel (**B**), left PV common trunk; and panel (**C**), right accessory pulmonary vein.

**Figure 2 jimaging-08-00300-f002:**
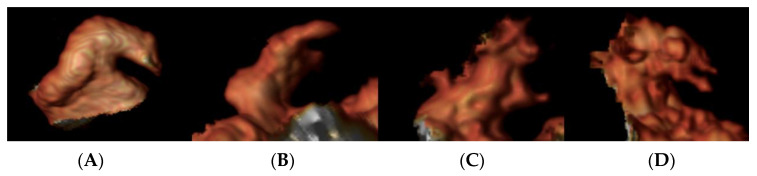
Left atrial appendage morphology on CMR images: Panel (**A**): chicken wing; panel (**B**): wind sock; panel (**C**): cactus; and panel (**D**): cauliflower.

**Figure 3 jimaging-08-00300-f003:**
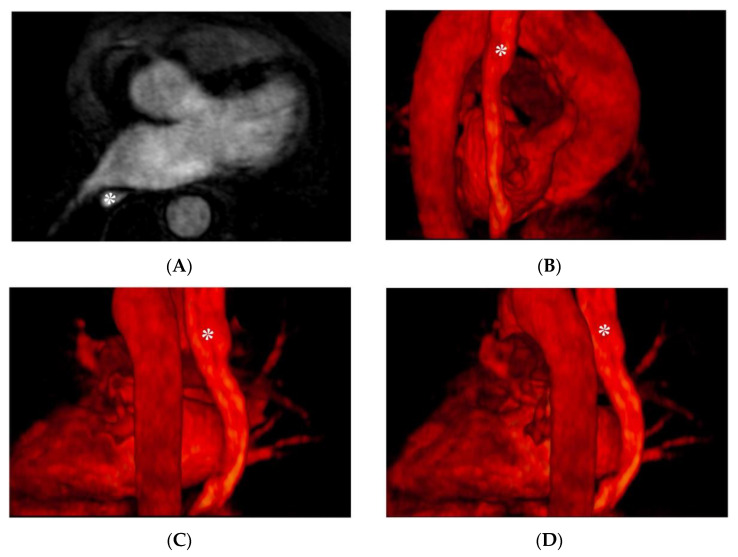
CMR images depicting the anatomical relationship between the esophagus (indicated with the asterisk): Panel (**A**): axial plane, angiographic CMR acquisition; panel (**B**–**D**) 3D volume rendering.

**Figure 4 jimaging-08-00300-f004:**
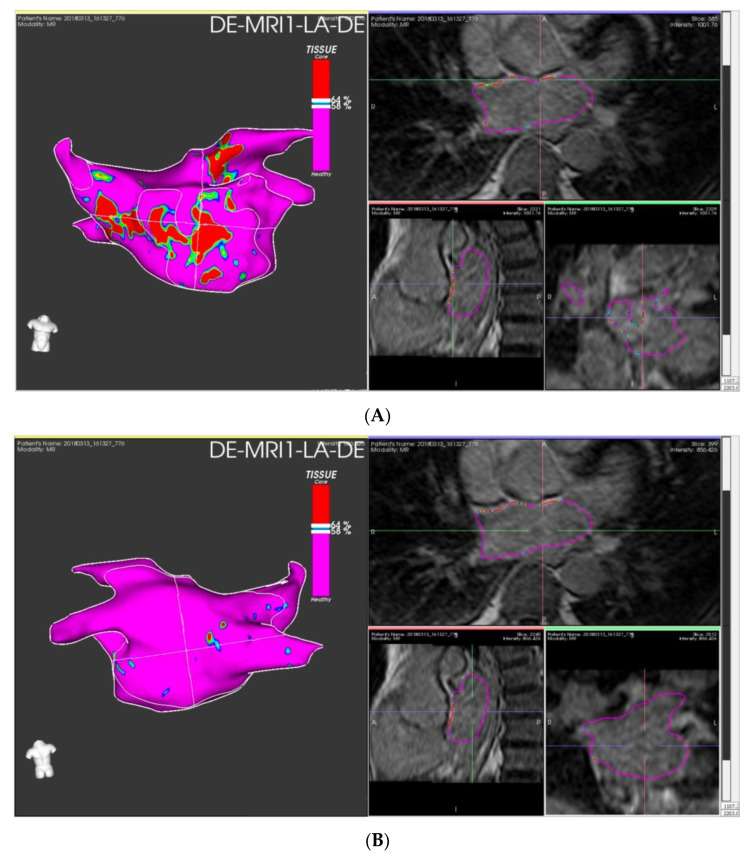
Left atrial fibrosis staging with LGE-CMR images elaborated with ADAS 3D^TM^ (ADAS3D Medical, Barcelona, Spain) before catheter ablation: Panel (**A**) anterior LA wall; panel (**B**) posterior LA wall. Estimated fibrosis: 12%, Utah stage II.

**Figure 5 jimaging-08-00300-f005:**
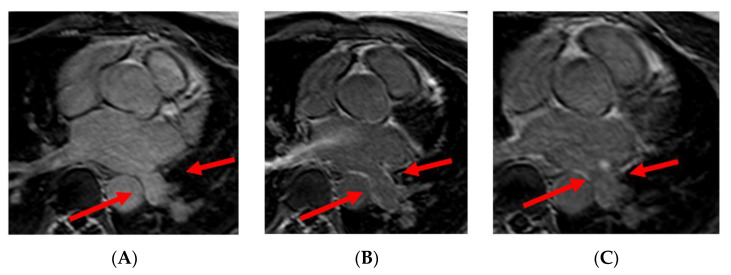
Panel (**A**): preprocedural LGE; postprocedural LGE at 24 h (panel (**B**)) and 30 days (panel (**C**)) after cryoablation. Red arrows indicate the pulmonary vein ostia.

**Figure 6 jimaging-08-00300-f006:**
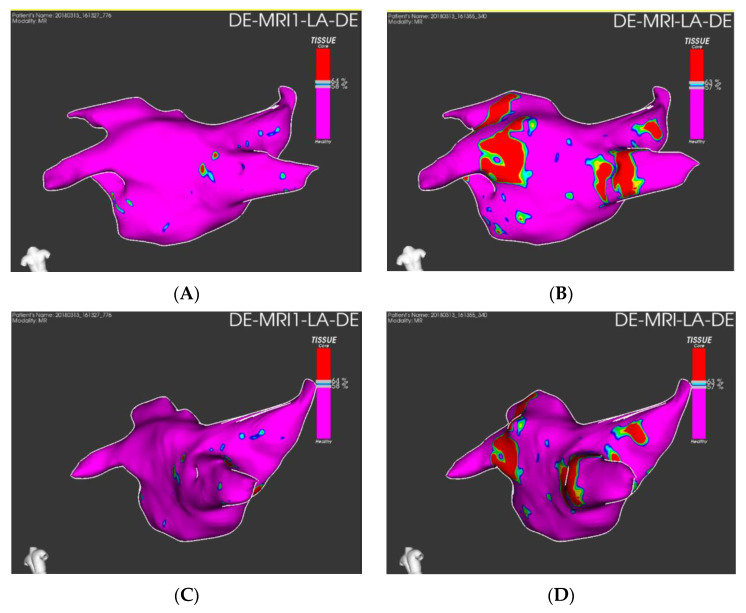
Quantification of preprocedural fibrosis (panel (**A**,**C**) and postprocedural fibrosis (**B**,**D**) on LGE-CMR images on ADAS 3D^TM^ (ADAS3D Medical, Barcelona, Spain).

**Figure 7 jimaging-08-00300-f007:**
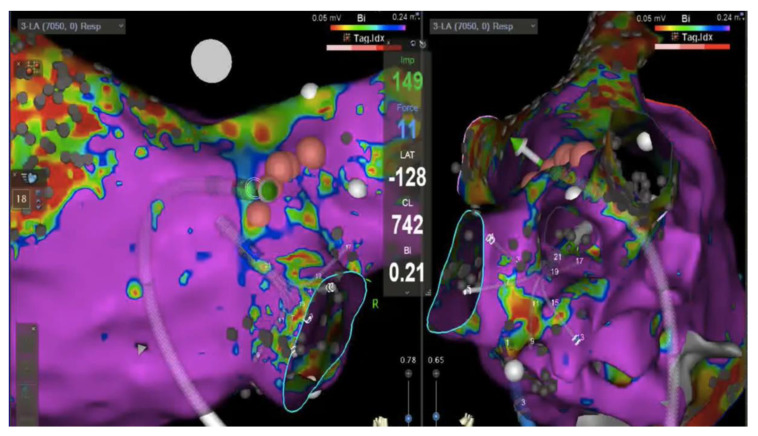
Hybrid imaging: Electoanatomic (Carto v7, Biosense Webster) voltage map of the left atrium (healthy tissue >0.24 mV, in violet); posterior view on the left and right lateral view on the right showing multipolar mapping catheter in the right inferior pulmonary vein antra. The ablation catheter (green vector) is pointing toward the posterior antrum of the vein with a real time 11 g contact force (value reported in light blue in the center of the image). Pink dots represent ablation sites.

**Figure 8 jimaging-08-00300-f008:**
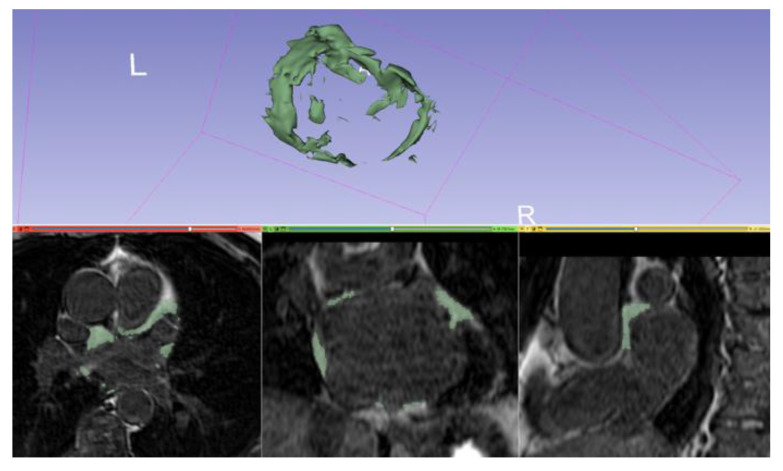
Manual segmentation of periatrial fat and volume rendering on a 3D IR spoiled gradient echo LGE acquisition with 3DSlicer (slicer.org, accessed on 10 June 2022).

**Table 1 jimaging-08-00300-t001:** Most used cardiac magnetic resonance sequences for atrial evaluation.

Protocol	Sequences	Key Points
Minimal protocol		
Survey	bSSFP	Images used to orientate subsequent acquisitions
Contrast enhanced MR angiography	T1 weighted spoiled gradient-echo sequence	To evaluate pulmonary vein anatomy, atrial size, presence of thrombus, esophagus course
Additional sequences		
Cine images	bSSFP	To evaluate LA/LAA volume, ejection fraction, strain
T1 mapping	MOLLI, ShMOLLI, STONE, SASHA, SAPPHIRE	To evaluate atrial wall native T1
Dixon chemical shift imaging	3D navigator gated mDIXON	To quantify pericardial/periatrial fat
T2 weighted imaging	T2 STIR, T2 TSE	To evaluate post procedural atrial and/or esophageal oedema
LGE	Inversion recovery gradient-echo (either 2D, 3D, single-shot, or phase-sensitive), breath-hold, or navigator gated	To quantify atrial fibrosis, to evaluate esophageal thermal injury

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
