# Peer review of "Role of Cardiovascular Magnetic Resonance in the Management of Atrial Fibrillation: A Review"

_2313-433X, 2022, doi:10.3390/jimaging8110300_

Round 1

Reviewer 1 Report

I congratulate the authors for good presentation of data although major clarity is needed on the "real world" application of CMR in this contest.

Author Response

Response to Reviewer

Original Manuscript ref.: jimaging-1819472

We would like to express our sincerest gratitude to the Reviewer for the insightful and constructive review of our manuscript. It is our opinion that we were able to increase the quality of the manuscript following his suggestions. Below, you can find a point-by-point response to his remarks.

Kind Regards,

The Authors

The Reviewer’s questions were highlighted in bold and our answers in italics.

Reviewer's comments:
Reviewer #1: 
However, there is a lack of presentation of data with comparison to the other imaging modalities such es echo and CT, which is essential in the clinical practice. I would appreciate a further discussion on the timing and indication of performance of CMR in AF patients (when and if CMR? Is the use of CMR supported by current guidelines?).

We thank the Reviewer for the kind remarks.

As suggested, we added a section in paragraph 4.1 reporting indications and timing for atrial LGE-CMR according to 2020 ESC guidelines.

We decided not to compare CMR to other imaging methods such as CT or US because it would have been beyond the scope of our paper which is focused only on the role of CMR for the management of atrial fibrillation.

Reviewer #1: 
In the 3 chapter I would suggest presenting a table which summarizes the CMR protocol for AF planning, with both standard and additional sequences and key points at each sequence.

As suggested, in chapter 3 we added a table summarizing the main CMR sequences used in the setting of catheter ablation procedures of atrial fibrillation.

Reviewer #1: 
Concerning the role of LGE-CMR in AF, authors should add a paragraph where major limits of atrial LGE are presented.

As suggested, we added a section in paragraph 4.1in which we discuss the limitations of LGE CMR reported in literature and we expose the reasons why it is not used in routine clinical practice.

Reviewer 2 Report

An up-to-date and comprehensive review overall. There are quite a few language corrections required. Some points need (more) references.

Author Response

Response to Reviewer 2

Original Manuscript ref.: jimaging-1819472

We would like to express our sincerest gratitude to the Reviewer for the insightful and constructive review of our manuscript. It is our opinion that we were able to increase the quality of the manuscript following his suggestions. Below, you can find a point-by-point response to his remarks.

Kind Regards,

The Authors

The Reviewer’s questions were highlighted in bold and our answers in italics.

Reviewer's comments:

Reviewer #2: 
Minor language revision suggestions.

We thank the Reviewer for the kind remarks.

As suggested, we made the language revisions requested throughout the paper.

Reviewer #2: 
Request for more references.

As suggested, we added the requested additional references throughout the paper.

Reviewer #2: 
Line 55: Consider revising this sentence.
As suggested, we revised the sentence and added further information to explain in a more precise way the origin of atrial ectopic firing from the pulmonary veins.

Reviewer #2: 
Line 422: Consider revising this sentence (maybe using a list) so that the reader understands the steps easier.

As suggested, we revised the sentence dividing it in a list making it easier to understand the different steps which take place in the electrophysiology lab.